# 3D Scaffolds to Model the Hematopoietic Stem Cell Niche: Applications and Perspectives

**DOI:** 10.3390/ma14030569

**Published:** 2021-01-26

**Authors:** Ada Congrains, Juares Bianco, Renata G. Rosa, Rubia I. Mancuso, Sara T. O. Saad

**Affiliations:** Hematology and Hemotherapy Center, Hemocentro-Unicamp, Campinas, São Paulo 13083-878, Brazil; juares@gmail.com (J.B.); renata.giardini@gmail.com (R.G.R.); rubiaimancuso@gmail.com (R.I.M.); sara@unicamp.br (S.T.O.S.)

**Keywords:** 3D culture, bone marrow niche, HSC, scaffold

## Abstract

Hematopoietic stem cells (HSC) are responsible for the production of blood and immune cells during life. HSC fate decisions are dependent on signals from specialized microenvironments in the bone marrow, termed niches. The HSC niche is a tridimensional environment that comprises cellular, chemical, and physical elements. Introductorily, we will revise the current knowledge of some relevant elements of the niche. Despite the importance of the niche in HSC function, most experimental approaches to study human HSCs use bidimensional models. Probably, this contributes to the failure in translating many in vitro findings into a clinical setting. Recreating the complexity of the bone marrow microenvironment in vitro would provide a powerful tool to achieve in vitro production of HSCs for transplantation, develop more effective therapies for hematologic malignancies and provide deeper insight into the HSC niche. We previously demonstrated that an optimized decellularization method can preserve with striking detail the ECM architecture of the bone marrow niche and support HSC culture. We will discuss the potential of this decellularized scaffold as HSC niche model. Besides decellularized scaffolds, several other methods have been reported to mimic some characteristics of the HSC niche. In this review, we will examine these models and their applications, advantages, and limitations.

## 1. Introduction

Hematopoietic stem cells (HSC) are rare, self-renewing, multipotent cells that are responsible for maintaining the blood and immune cells supply through a process called hematopoiesis. HSCs undergo several fate decisions including self-renewal and differentiation that are key for hematopoietic homeostasis. Evidence demonstrates that HSC fate is dependent on the surrounding environment, the specialized microenvironments in the bone marrow called niches. The bone marrow niche comprises many cellular, biochemical and physical components and the interaction between them defines specific stem cell functions [1,2,3,4]. Despite the clear importance of the niche in HSC function, most experimental approaches to study human HSCs malignization, drug efficacy, hematopoiesis, HSC aging are based on conventional 2D culture.

Different niches are associated with defined anatomic structures and distinct residing cells in the bone marrow and HSC subpopulations with different proliferation and differentiation potential [1,2,3,4]. A population of self-renewing HSCs remain in a quiescent state, protected from genotoxic insults and exhaustion [5], while committed progenitors mobilize to other niches and finally into circulation [1]. This balance of differentiation and quiescence is essential to maintain homeostasis and health. Strong evidence proposes a decisive role of the niche in these fate decisions [1,2,3,4].

HSCs are probably the most clinically used adult stem cells. These primitive progenitors are responsible for the reconstitution of the hematopoietic system upon bone marrow ablation and transplantation [6]. In addition to stem cell transplantation, HSCs have been extensively studied due to their role in hematologic disease. Disfunction of HSCs is responsible for leukemogenesis [7], immune deficiencies [8], immuno-senescence [9], and several hematologic diseases. In addition, the hematopoietic system is highly sensitive to radiotherapy and chemotherapies [10]. Hematologic toxicity is a common and life-threatening complication that limits the maximum doses administrable for several drugs [10].

Most efforts to study the bone marrow niche have been focus on the understanding of HSC behavior and hematopoiesis. Besides hematopoietic cells, the bone marrow also hosts another important stem cell lineage, mesenchymal stem cells (MSC). MSCs can differentiate into different cell lineages in vitro and in vivo, modulate immune response and inflammatory pathways in other cells [11]. Changes in the MSC physiology are associated with numerous health conditions [12,13]. Several mesenchymal stem cell therapies are currently under trial for numerous applications, including bone repair after fracture, Acute graft-versus-host disease (aGVHD), cardiovascular repair and liver disease [11,12]. Although the focus of this review is HSC niche models, some 3D models of the bone marrow used to study the MSC physiology will be also discussed.

Despite advances in the understanding of the role of the functional and structural components of the HSC niche, there is no ex vivo model that can faithfully reproduce the in vivo HSC niche homeostasis, long-term stem cell maintenance and complex cellular interactions of the niche. Recreating artificially the extracellular matrix (ECM) composition and intricated structure of stem cell niches is a bioengineering challenge for the decades to come. To face this problem today, decellularized ECM scaffolds from tissues and organs came as a short-cut that was used effectively as in vitro model and organ engineering platform [14,15,16,17,18,19]. To study the HSC niche or model a pathological condition of the bone marrow, a system that could faithfully reproduce the physical architecture, chemical and cellular components of HSC niches seems ideal. However, for some applications that require easy fabrication, scale up, and uniformity, only a few essential elements of the niche need to be introduced, and synthetic scaffolds are more attractive.

Several models successfully recapitulated some aspects of the bone marrow niche and contributed to HSC ex vivo expansion and understanding of HSC physiology. In this review, we will overview the current knowledge of elements of the HSC niche and describe up-to-date 3D culture HSC niche models.

## 2. Hematopoietic Stem Cells and Their Niche

Hematopoiesis occurs primarily in the bone marrow and relies on the multi-potency and self-renewal capacity of the hematopoietic stem cell to take place throughout life. Remarkably, the microenvironment must provide the appropriate signals, soluble cytokines, tridimensional cues and cellular interactions for long-term maintenance and differentiation of HSCs into different blood lineages. All these factors interact to regulate the balance between self-renewal and differentiation that is critical to maintain the stem cell pool and prevent malignant proliferation.

The concept of an HSC niche was first proposed in 1978, as the surrounding environment of HSCs. Although the anatomic location of the HSC niche has been subject of controversy, evidence points to the existence of an arteriolar niche, endosteal niche, and peri-sinusoidal niche [1,2,3,4,20,21]. In their niches, HSCs are able to maintain their self-renewal capabilities and evade differentiation in response to the signals received from their microenvironment. While in their natural microenvironment HSCs maintain a life-long self-renewal capacity, in culture these primitive progenitors rapidly differentiate and ex vivo long-term maintenance of HSCs has remained challenging.

To recognize the need for an accurate HSC niche model, we need to understand the complexity of the HSC niches and the cellular interactions that maintain homeostasis. The bone marrow is a structurally complex organ hosting subpopulations of hematopoietic progenitors and non-hematopoietic cells, vascularized, innervated and confined by the bone. The inner surface of the bone, called endosteum, is lined by a thin cellular layer of osteoblasts and osteoclasts. The endosteal “osteoblastic” niche was the first putative HSC niche [21]; however, the common assumption of direct contact of HSCs with osteoblasts have been disputed by recent evidence [3,22]. The endosteum is an important lymphopoietic site, and osteoblast secretion of CXCL12 is crucial for lymphoid differentiation [1]. While C-X-C motif chemokine ligand 12 (CXCL12) expressed by osteoblasts seems to be responsible for early lymphoid progenitor maintenance, CXCL12 expressed by endothelial and stromal cells influences maintenance of HSCs [1].Osteoblasts are known to regulate HSC proliferation and erythroid differentiation by Osteopontin [23] (OPN) and erythropoietin [24] (EPO) production. However, evidence shows that less than 20% of HSCs are in direct contact to the endosteum [22] and most recent evidence suggests that the niche of HSCs is primarily perivascular, remarkably around the arterioles and sinusoids [1,2,3,20,25].

Arterioles, which are close to the endosteum [3], are important HSC sites and are essential for HSC quiescence and maintenance [3,25]. The arteriole niche harbors several populations of stromal cells (particularly important, leptin receptor-expressing, CXC-chemokine ligand 12 (CXCL12)-abundant reticular (CAR) perivascular stromal cells and neural–glial antigen 2 (NG2) periarteriolar cells), endothelial cells, sympathetic nervous system (SNS) nerves and non-myelinating Schwann cells. They all contribute with chemical signals to HSC homeostasis. CAR cells are the main sources of CXCL12, they localize surrounding endothelial cells in the sinusoids and arterioles and are in direct contact to HSCs [26]. Schwann cells together with sympathetic nerves in the arteriole induce quiescence of HSCs through transforming growth factor beta (TGF-b) activation and maintain direct contact with a considerable number of HSCs [27]. Localization of the primitive hematopoietic stem cells in specific niches is subject of controversy, probably due to the different markers and limitations of the methods used to identify long-term HSCs. Although strong evidence supports that quiescent HSCs localize away from the bone, mainly around the sinusoids [28], another line of evidence showed that primitive HSCs, or at least a subpopulation, localize near the arterioles, and proximal to the endosteum [20]. In line with these findings, NG2^+^ stromal cells in the arteriole niche have been associated with lymphoid biased HSCs and megakaryocytes in the sinusoids have been linked to myeloid-biased HSCs [29]. Interestingly, myeloid differentiation bias is associated with immunosenescence, aging [30] and cancer [31]. Moreover, megakaryocytes in the sinusoidal niche has been shown to regulate HSC quiescence through C-X-C motif ligand 4 (CXCL4), TGF-b and expansion under stress through fibroblast growth factor 1 (FGF1) [4,32]. A summary of these interactions is presented in Figure 1.

Chemical and physical characteristics of the bone marrow have effects in HSC behavior. Oxygen tension, which is considered central to HSC quiescence, is not uniform throughout the bone marrow. Direct measurement of oxygen pressure identified gradual differences, and contrary to common thought, endosteal regions are the most oxygenated and perisinusoidal regions the most hypoxic [33]. Bone marrow stiffness, which regulates stem cell, MSC and HSC behavior [34,35], is also different across the marrow: the rigid bone is >10^6^ KPa, the osteoblastic matrix of the endosteal surface is much less rigid (>30 kPa), while the central marrow is soft (0.3 kPa) [35,36]. Material stiffness seems to be a particularly important consideration for 3D culture scaffolds [37,38,39]. Interestingly, scaffold rigidity induces greater HSC adhesion and migration and has been associated with a myelo-erythoid bias in vitro, while softer matrices promote granulocyte differentiation [40]. In addition, osteogenic differentiation of stromal cells also is induced by substrate stiffness [41], which could indirectly contribute to the formation of an in vitro endosteal niche.

In addition to cellular elements, the extra-cellular matrix is an important active element of the niche. It is clear that changes in the physical characteristics and composition of the ECM affect stem cell function [42]. The constituents of the ECM may comprise more than a 200 proteins [43], including collagens, proteoglycans, Laminin, fibronectin, elastin and ECM-associated growth factors, cytokines and ECM remodeling enzymes. These elements are continuously being synthesized, modified, and secreted by the cellular components they support. These components are dynamically interlinked, for instance, cytokines and growth factors are often sequestered in the ECM, regulating their availability and matrix metalloproteinases are remodeling the ECM in which they are embedded [44]. Using state-of-the-art imaging and proteomics analysis, Mayorca et al, demonstrated that cancer progression drives dramatic ECM remodeling that includes changes in composition and structure [45]. The ECM is actively connected to the cell through integrins, syndecans, and other receptors. Fibronectin, laminin, and collagen are known to be required for migration and proliferation of HSCs through a common integrin-binding domain, termed RGD [40]. Unsurprisingly, introducing these anchorage peptides (RGD) in artificial scaffolds has shown superior performance in adhesion, repopulation and differentiation of several stem cells [46,47], including HSCs [48].

The bone marrow environment is not homogeneous and at least three distinct HSC niches have been extensively described: endosteal, arteriolar and sinusoidal. Each of these microenvironments has particular ECM characteristics, distinct cellular elements, oxygen tension and physical properties that regulate HSC physiology.

## 3. Applications of 3D Models of BM Niche for Cell Culture

### 3.1. Ex Vivo Expansion of HSC

Hematopoietic stem cell transplantation (HSCT) is the only potentially curative treatment for numerous hematologic disorders. Access to this life-saving procedure is limited by the availability of an HLA-matched donor, which could be as low as 16% in some ethnic groups [49]. Ex vivo expansion of hematopoietic stem cells in an artificial culture system would provide a therapeutic alternative for patients without available donors.

Evidence from serial transplantation in mice demonstrates that HSCs have extremely long-term self-renewal capacity in vivo [50]. However, long-term maintenance of HSCs ex vivo has proven challenging and in vitro expansion of repopulating HSCs have had limited success [51,52]. In this sense, in vitro HSC expansion systems should ideally mimic key microenvironmental characteristics of the bone marrow to achieve homeostasis and long-term maintenance of HSCs. Evidence suggests that signals existing in the tridimensional microenvironment of the cell are key to stemness maintenance and 3D culture systems are superior in maintaining pluripotency of several stem cells [53,54,55]. In addition to tridimensionality, chemical signals from other cellular elements of the niche and ECM components are also important and probably should be considered to establish maintenance and expansion ex vivo [3,4,26,27,56,57].

Probably expansion of HSCs is the most studied application for 3D culture of HSCs. Numerous synthetic 3D scaffolds have proven to provide a more favorable environment for HSC expansion and maintenance of pluripotency than bidimensional cultures [48,58,59,60,61,62], see Table 1 for a brief description. Although the use of 3D scaffolds for HSC expansion is attractive, cell recovery from the scaffold after potential expansion might be challenging depending of the scaffolding material. In the case of an ECM decellularized bone marrow (such as DeBM presented in Section 4.2), collagenase treatment could be an option for cell recovery; however, feasibility of this approach has not been tested. The combination of a bioreactor with a 3D scaffold could allow the release of HSCs into the liquid phase; however, in [61], the authors noted that the released cells showed differentiation commitment. Remarkably, they were able to recover HSCs from the solid phase using perfusion with collagenase in the bioreactor; however, viability of the recovered population was not stated [61].

Scaffold-less 3D culture approaches, such as cellular microspheres, have been shown to also enhance HSC expansion [55,63]. Fewer natural decellularized scaffolds have been described as bone marrow niche models in the literature [18,64]. Although these models might resemble better the natural microenvironment of HSC, no reports of their suitability for HSC expansion are currently available.

### 3.2. Bone Marrow Study Model

Traditional bi-dimensional culture systems fail to reproduce the behavior of the natural tissue. Evidence shows that monolayer culture ultimately modifies cellular phenotype and gene expression compared to tridimensional culture and natural tissue [65,66]. For decades, we have used cell monolayer or suspension cell culture models to understand cellular physiology. These culture systems dismiss completely the effect of the niche: tridimensionality, ECM-cell interactions and complex multi-cellular dynamic interplay.

Immortalized leukemia cell lines adapt very well to 2D culture and are useful models of disease. However, the study of leukemogenesis from human hematopoietic cells to leukemic stem cells depends on the study of primary HSCs. Maintaining the natural phenotype of HSCs in a traditional culture system requires demanding conditions and even in optimal conditions, changes in the expression profile and phenotype have been reported [67]. In vitro 3D models of the BM niche are valuable tools to study HSCs and unravel changes that lead to malignization. For instance, a humanized gelatin-based porous scaffold seeded with MSC was used successfully as model In vitro and in vivo. Moreover, it allowed successful xenotransplantation of primary cells from leukemia patients that showed no engraftment using traditional methods [68], allowing studying these cells in vivo.

An ex vivo HSC microenvironment also allows studying hematopoiesis in homeostasis and disease. A collagen scaffold-based 3D culture system allowed to study migration and differentiation bias of HSCs in different compartments of their system [69]. An artificial thymic organoid (ATO) system allowed the study in vitro of T-cell differentiation. Differentiation from HSC to T cells exhibiting mature naive phenotypes recapitulated in vivo differentiation and was much more efficient in the ATO system than in conventional 2D systems [70].

In addition, studying the HSC niche requires a tridimensional setting. Most of our understanding of the HSC niche comes from mouse models and microscopic analysis of the murine bone marrow. Although many of the insights obtained from these models have proven to extrapolate in the human HSC niche, there are fundamental differences between mice and human hematopoietic system [71]. In this sense, 3D culture systems are instrumental to study characteristics of the human HSC niche. ECM is produced by cells in vitro, and this production of matrix can be exploited to obtain natural scaffolds that resemble natural characteristics of the tissue ECM. An ECM scaffold produced using this approach revealed specific characteristics of the ECM produced by osteoblasts, endothelial and mesenchymal cells and assessed their performance as platforms for MSC and HSC culture [72]. Unquestionably, the development of a realistic ex vivo model of the human bone marrow would be an unprecedented tool to study HSCs, MSCs and niche interactions in a natural context.

### 3.3. Large Scale Drug Testing Platforms

Drug development is a lengthy and expensive process that encompasses several stages from target identification to clinical trials. Despite the tremendous effort invested in pre-clinical drug testing, phase II and III clinical trials have very high attrition rates [73]. Remarkably, more than half of failures are due to a lack of efficacy [73], suggesting severe limitations in current pre-clinical models.

High-throughput screening of small compound libraries for identification of potential therapeutic compounds is an essential step in the drug development process. The majority of cell-based high-throughput screening approaches rely on well-characterized cell lines in bidimensional cellular culture systems. These systems do not allow for the influence of surrounding cells, molecular diffusion gradients and physical characteristics of the cell environment. All these elements are relevant to cellular behavior and treatment efficacy. Stromal cells are key to drug resistance and drug efficacy is reduced when malignant cells are associated with stromal cells [74]. Diffusion of candidate drugs to the target cell, oxygen, nutrient diffusion, and cellular migration are known variables that affect drug delivery and efficiency [75]. The use of synthetic scaffolds or decellularized tissue present a physical barrier that will affect the diffusion of drugs, cytokines and/or other substances and alter cellular migration. In addition, biochemical characteristics of the scaffolding material or method used for decellularization (in the case of decellularized tissue) may affect the barrier properties and general performance of the scaffold. New tools and methods that take into account these factors are needed to increase precision in the assessment of drug candidates.

On the downside, high-throughput screening (HTS) assays are, in most part, not compatible with three-dimensional culture systems. Traditional imaging methods have limited optical access into tridimensional structures, co-culture of multiple cell types and irregularity of the natural niches are factors limiting HTS application of 3D culture. In addition to imaging, gene expression has been used to assess phenotypic changes in HTS approaches in the context of hematologic disease [76]. Tridimensional culture will also present challenges for gene expression-based high-throughput screenings, depending on the scaffolding material, recovery of cellular material might be problematic and successful implementation will also depend on uniform seeding and adhesion.

Customizable synthetic polymers and natural scaffolding materials have significant advantage for HTS, and several efforts to use 3D scaffolds in HTS have shown feasibility. Matrigel, which is a complex mixture of natural proteins from the ECM, is suitable for nanoliter-scale 3D culture ‘on chip’, allowing the evaluation of more than 500 conditions for HTS [77]. This approach also supported superior proliferation of a neural progenitor cell line compared to 2D culture. In another example of HTS application, a 3D porous polystyrene scaffold plated in regular plates was used in neural progenitor culture. The authors demonstrated that cells clustered, proliferated and differentiated in a manner similar to neuro-spheres, considered reference of in vivo behavior [78]. Scaffolds using artificial polymers are highly homogeneous, easier to fabricate and scale into 96-well plates to be compatible to most high-throughput screening equipment; however, these platforms lack many key features of the natural microenvironment. We previously demonstrated that a decellularized bone marrow scaffold preserves histologic characteristics of the bone marrow niche and supports HSC adhesion and growth [18]. Similar decellularized lung scaffolds have been proposed to be a suitable platform for high-throughput screening analysis [79]. However, to our knowledge, no HTS assay using a decellularized tissue scaffold has been developed to quantitively measure a specific phenotype. Probably one of the most challenging obstacles is the irregular topology of natural tissue that complicates uniform adhesion of seeded cells and subsequent quantitative interrogation of phenotypic changes. Although the use of 3D culture systems that recapitulate niche complexity in drug discovery and repositioning is very promising, many limitations need to be addressed before 3D culture is widely used for high-throughput and automated screening.

## 4. Current 3D Models of the Bone Marrow Niche

A wide variety of materials are currently used in the production of artificial scaffolds for cell culture applications. Scaffolds are usually porous and polymeric structures, which allow cell infiltration and support cell growth, and easily customizable. Despite their wide use in cell culture, and their advantages, they lack the complexity of the native extra-cellular matrix and present problems of bio-compatibility [80]. Natural scaffolds obtained from decellularized tissues maintain the tridimensional structure of the native tissue and the only scaffolds that have been used to recreate functional organs [14,15,16,19,81]. Upon seeding into the decellularized scaffold, cells are able to recognize their native cellular microenvironment and these sole cues lead to proliferation and differentiation [17]. Moreover, extra-cellular matrix components, such as collagen and fibronectin are highly conserved proteins which explained that decellularized tissues present low cytotoxicity and immunogenicity even between different species [82].

Several attempts to reproduce the bone marrow microenvironment using 3D culture models have been reported. Early attempts of HSC tridimensional culture using scaffold-free 3D culture have been reported. Bioreactors were used to increase cell-cell interactions and optimize availability of cytokines and nutrients [83,84]. This type of scaffold-free systems had limited success in HSC expansion. A hanging drop system was implemented using co-culture of MSC and HSC; however, it presented problems of migration and it did not succeed in expanding HSCs [85]. Culture conditions previously associated with hematopoietic stemness maintenance such as hypoxia [86,87] have been incorporated in 3D models. Gradient hypoxia in an hydrogel scaffold produced distinct niches, which were abolished in an hypoxic environment [88]. Dynamic and steady culture conditions were assessed, using a sophisticated 3D culture system, in as attempt to mimic the natural blood flow in the HSC niches; however, no significant differences in HSC maintenance were observed [89]. We will describe the 3D culture methods recently develop for HSCs and other niche cells. A brief summary of these methods described in the literature is presented in Table 1.

### 4.1. Decellularized 3D Model of the Bone Marrow Niche

Natural bio-scaffolds derived from decellularized tissue have been prepared from a variety of tissues [15,16,17,19,82,95]. Decellularized 3D scaffolds are the most complex and mimic more realistically the cellular environment, and the only platform that, so far, could support whole organ re-engineering [15,16,96]. These tissues/organs can be harvested from various species including pigs, cows, horses, and humans [97]. Unlike available synthetic scaffolds, natural decellularized scaffolds possess the native complex architecture and composition that are essential for mediating cellular responses. These unique properties ultimately facilitate integration into the host tissue after implantation and elicit less immune response toward the ECM components upon adequate decellularization treatment [98,99]. Decellularization of extracellular matrix produced by cells in vitro has also offered a simplified alternative to mimic the in vivo microenvironment of cells [57,100].

Several optimization strategies have been implemented to achieve sufficient preservation of the native structural and functional components for successful decellularization [101,102]. Furthermore, thorough removal of any cellular and DNA residues is also critical in preventing adverse immune reactions especially in xenogenically sourced ECM [102,103].

The decellularization starts with physical methods used to lyse cells, between all the methodologies the Freeze/thaw cycles are the most commonly used. The rapid changes on the temperature can freeze the intracellular fluid and the ice crystals formed by this process disrupt the cellular membranes causing cell lysis [104]. During the freeze/thaw cycles a meticulous control of the temperature is necessary to avoid that the formation of ice crystals affect the extracellular matrix ultrastructure. However, physical methods are not enough to extract all the cell content and additional chemical and enzymatic methods are necessary. It is common the use of chemical treatments with detergents and/or enzymatic digestion-based approaches, or a combination of salt solution rinses with enzyme digestion/detergent rinses. Sodium dodecyl sulfate (SDS) was commonly used to complete the decellularization especially in larger tissue, where other milder detergents were not effective. However, Flynn et al. pointed out that the use of SDS damage the vascular architecture of the tissue, complete removal is problematic and its use interfere with subsequent enzymatic digestion of DNA and RNA [105]. Another study also suggested that use of detergents in the decellularization process affect the biomechanical characteristics of the scaffold [106]. Several recent methods are achieving successful decellularization without the use of detergents [18,106,107].

Our group developed a bone marrow decellularized scaffold (DeBM) using a freeze-thaw-based method excluding the use detergents to prevent degradation of the ECM and the presence of potentially cytotoxic chemical residues. Brief description of the main steps of the method in Figure 2, details of the protocol in reference [18].

We demonstrated that the decellularized scaffold is cell-free and is capable of supporting HSCs adhesion and proliferation [18]. To our knowledge, only two decellularized bone marrow scaffolds have been described in the literature [18,64]. Hashimoto et al. used high hydrostatic pressure for decellularization of a porcine bone/bone marrow. This scaffold was tested as platform of MSC culture and osteogenic differentiation. The applicability of this scaffold for in vitro HSC culture was not assessed; however, recruitment of HSCs was observed in vivo after subcutaneous implantation [64]. The focus of the study was to develop a platform for osteogenic differentiation, and probably that is why most microscopic evaluations of the integrity of the natural structures of the scaffold were focused on the cancellous bone rather than the delicate architecture of the bone marrow soft tissue.

On the other hand, the decellularization method implemented by Bianco et al. achieved detailed preservation to the level of individual cellular niches and intact vascular structures throughout bone marrow soft tissue, as seen in Figure 3. We previously reported the biocompatibility of this scaffold as culture platform for human HSCs, and immortalized HS5 mesenchymal stem cell line [18].

In recent, not published evaluation of the scaffold, we observed that primary human non-hematopoietic niche cells (CD34^-^ cells) also adhere to specific anatomic regions of the bone marrow scaffold as seen in Figure 4. Culture of non-hematopoietic stromal niche cells may functionalize the scaffold for HSC culture in future experiments.

There are several clear advantages of a decellularized bone marrow scaffold: availability of the raw material, no need for a priori knowledge of all relevant elements of the niche, no need for specialized equipment, and of course the preservation of the natural chemical and physical characteristics of the tissue. However, due to the large variability of ECM ultrastructure and composition through the bone marrow, experimental scaling up and reproducibility of cellular assays may be problematic. For instance, cellular adhesion and proliferation signals in the ECM might be different in the endosteal and the central region of the natural scaffold. Natural anatomical elements of the bone marrow may interfere with uniform seeding of cells in the scaffold, and particularly complicate automated approaches of quantification.

### 4.2. Other Models

#### 4.2.1. Synthetic Scaffolds

The advancement in design and fabrication of biocompatible materials for bioengineering applications, allowed the use of matrices or scaffolds in 3D culture. In general, even simple approaches of 3D culture using synthetic and natural polymers already showed superior performance than 2D culture in preserving the characteristics of HSCs ex vivo [58,59], see Table 1. The study by Ventura et al. compared Polycaprolactone (PCL), poly-lactic-co-glycolic acid (PLGA), fibrin and collagen as scaffolding materials for HSC expansion [58]. This study used umbilical cord (UC) CD34^+^ HSCs in co-culture with UC-MSC, as a result, all but PLGA scaffolds were able to support HSC expansion. The greater HSC expansion and most primitive phenotype was obtained in the fibrin scaffold [58]. Porous polyurethane (PU) was also used as scaffold to culture CD34^+^ HSCs. This model was able to maintain stemness and some degree of homeostasis demonstrated by the constant egress of differentiated cells from the system [59].

Since then, more sophisticated systems have been devised. Enhancements include addition of integrin anchorage peptides [48], cleavable site for biodegradation [90], use of newer biocompatible zwitterionic hydrogels [90]. Zwitterionic hydrogels have attracted great attention for a wide range of biomedical applications. A unique characteristic of zwitterionic materials is the resistance to non-specific protein adsorption (nonfouling) [108]. The nonfouling capability of this type of materials make them particularly biocompatible and suitable for clinical use, since protein adsorption in vivo could lead to coagulation or trigger inflammatory responses.

Another approach is the deposition of ECM onto the scaffold by stromal cells (i.e., MSCs, osteoblasts, endothelial cells) [61,62,68] as a preparation for culture of HSCs. Culture of stromal cells in some scaffolds induce the production of ECM elements that mimic the natural niche matrix [18,61,62,68,91], and this functionalization contribute to adhesion and natural responses in HSCs. In an example of this type of system, a ceramic scaffold was functionalized by deposition of ECM from osteoblasts and stromal cells in a bioreactor culture. After functionalization, HSCs were able to adhere specifically to the scaffold, maintain pluripotency and differentiate [61]. Interestingly, differentiated cells were released to the medium while more primitive cells were retained in the scaffold, resembling the natural cellular behavior in the bone marrow.

#### 4.2.2. Microspheroids/Organoids

Spheroid culture methods have been used with success to expand several adult stem cells [109,110]. In the case of HSCs, some microsphere methods have been reported. MSC-laden collagen microspheres were osteogenically differentiated, to induce deposition of natural ECM components, and subsequently decellularized, to use as scaffold [91]. The scaffold developed using this approach was superior to collagen-only microspheres to induce proliferation of HSCs and MSCs [91]. Cellular microspheres also form spontaneously without the use of scaffolds in non-adherent surfaces, and this type of culture systems have also been developed as alternative for HSCs culture [55,63]. Mononuclear cells from peripheral blood containing negligeable number of primitive HSCs were able to spontaneously self-organize in ‘hematospheres’ when cultured in non-adherent plates [55]. In these *hematospheres*, primitive HSCs increased dramatically in number, appearing as a very attractive method to expand transplantable HSCs from blood [55].

Another method used a nestin^+^ subpopulation of MSCs to produce cellular spheres (mesenspheres) [63]. Nestin^+^ cells, besides their self-renewal capacity and multipotency, support HSC maintenance. In fact, co-culture of HSCs with these *mesenspheres* expanded transplantable HSCs and increased in vivo engraftment of HSCs [63].

Pievani et al. took the spheroid culture a step further, developing a functional organoid from chondroid pellets. Pellet culture of cord blood fibroblasts were differentiated into cartilaginous tissue in vitro. After subcutaneous implantation in mice, the chondroid rudiments remodeled into ossicles with resemblance of the bone/bone marrow architecture [92]. Vascular structures recreating sinusoids and hematopoietic tissue of erythroid, myeloid and megakaryocytic lineages were detected after 8 weeks. Remarkable, these organoids supported the engraftment of human HSCs and hematopoiesis in vivo [92].

#### 4.2.3. 3D Printing

Recent advancements in three-dimensional (3D) bioprinting technology and development of biomimetic materials, allow the deposition of a combination of biomaterials and cells to produce complex tridimensional living tissue-like constructs.

Recently developed methods in 3D bioprinting have produced astounding reconstruction of complex tissue structures using hydrogels, collagen and other materials [111,112,113]. 3D bioprinting is probably the most promising technology for 3D stem cell culture, tissue engineering and clinical applications. However, some major technical challenges need to be overcome such as resolution to recreate microscopic structures, limited capacity to reproduce intricated vascular networks, gelation issues, cellular viability, among others [114].

Some 3D culture approaches for HSCs have used 3D printing to mimic the bone marrow niche. A 3D printed hydrogel mesh loaded with MSCs was able to support proliferation of HSCs and proved to be a feasible co-culture scaffold. This mesh was superior than conventional 2D co-culture to support expansion of HSCs [93]. However, this 3D scaffold did not reproduce any anatomical structure of the bone marrow niche.

Another more sophisticated attempt to mimic the bone marrow niche used a two-compartment approach [94]. A 3D printed calcium phosphate cement (CPC) scaffold was seeded with MSCs and subsequently differentiated to osteoblasts to mimic the endosteal niche. A Matrigel loaded with endothelial and MSCs aiming to emulate the perivascular niche was incorporated into the rigid CPC scaffold to allow interaction and migration of cells between compartments. This system supported CD138^+^ primary myeloma cells proliferation and was used to model myeloma pathophysiology [94]. Applicability of this 3D model for HSC culture was not assessed.

## 5. Conclusions

In the previous sections we have discussed the complexity of the bone marrow niches and the relevance of several elements of the niche in HSC physiology. The reconstruction of all these elements artificially into a functional bone marrow remains elusive with current technologies. However, decellularization of tissues and organs may provide an alternative to bypass these shortcomings. We presented a decellularized bone marrow model developed in our laboratory as a promising scaffolding material for HSC culture, expansion, and drug testing platform. In addition, we revised several tridimensional models, materials and strategies used to emulate some important characteristics of the HSC microenvironment.

The final objective of these models would be to provide the necessary signals to cells in order to recapitulate the bone marrow niche. A model that successfully mimics the bone marrow microenvironment would have various applications in the research, clinical and pharmaceutical fields. It would be a tremendous tool to study hematologic diseases and normal hematopoiesis in a realistic environment; for expansion of HSCs for transplantation, and as platform for drug screening approaches.

## Figures and Tables

**Figure 1 materials-14-00569-f001:**
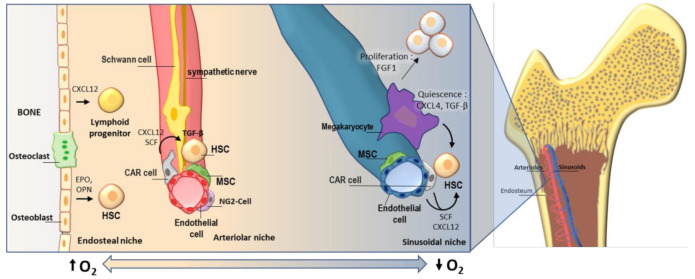
A graphical overview of the bone marrow components and cellular interactions taking place in the bone marrow niche.

**Figure 2 materials-14-00569-f002:**
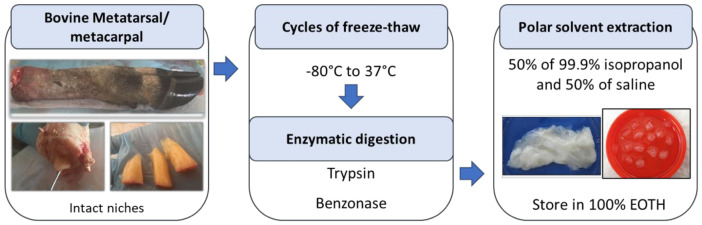
Graphical summary of the decellularization process. More details of the protocol in Bianco et al. [18].

**Figure 3 materials-14-00569-f003:**
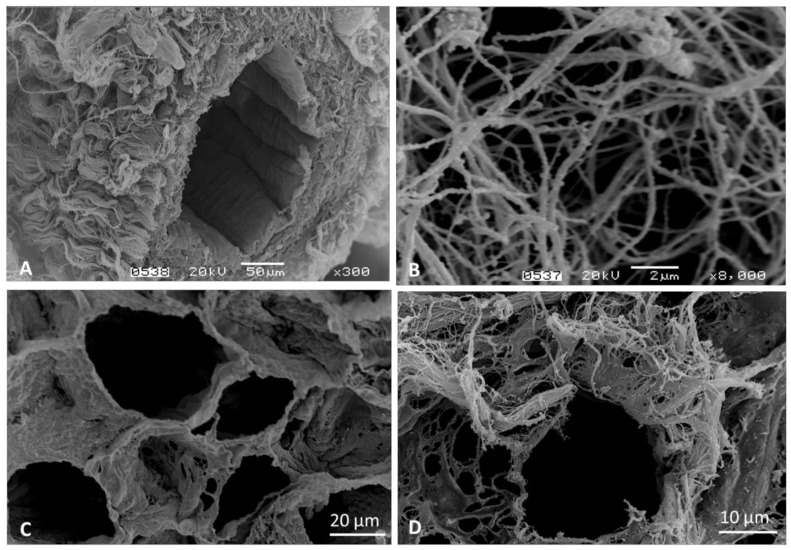
Scanning electron microscopy imaging of the bovine decellularized bone marrow. (**A**) Intact vascular structure, (**B**) Reticular fibers in connective tissue, (**C**) Adipose tissue ECM, preserved cellular niches, (**D**) Individual cell niche. The images are part of the Hematology and Hemotherapy Center archive of the characterization of the decellularized bone marrow (DeBM) [18]; however, these images have not been published elsewhere.

**Figure 4 materials-14-00569-f004:**
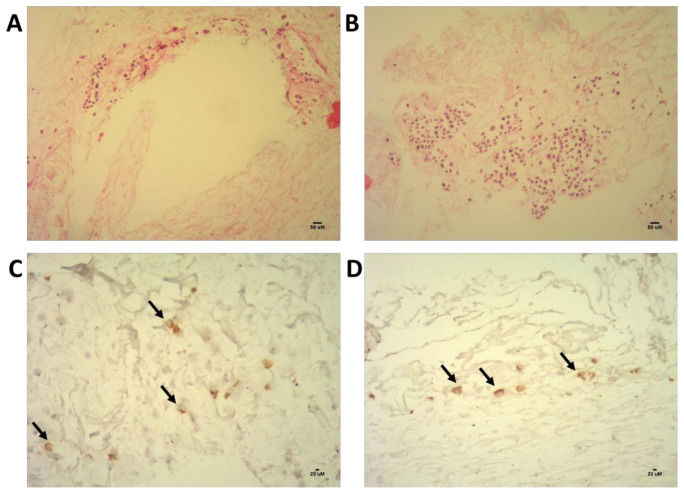
Non-hematopoietic niche cells (CD34^−^ cells) from a human donor adhered to specific regions of the bone marrow niche model and proliferated in the decellularized scaffold (panels **A**,**B**). Lower panels (**C**,**D**) showed the CD90^+^ cells (marker of mesenchymal cells). The images are part of recent characterization of the decellularized bone marrow (DeBM) and have not been published elsewhere.

**Table 1 materials-14-00569-t001:** 3D culture systems developed to mimic the bone marrow niche.

System	Description	Main Results
Synthetic scaffolds		
Poly (lactic-co-glycolic acid) (PLGA)	PLGA is a biocompatible and biodegradable material.	It does not support CD34^+^ cells growth [58].
Polycaprolactone (PCL)	elastic mechanical properties and slow degradation rate	Supports CD34^+^ adhesion and proliferation [58].
Polyurethane (PU)	PU is a polymer with attractive mechanical properties and biocompatible.	Supports Cd34^+^ proliferation, differentiation and egress [59].
Non-woven polyester fiber/polypropylene mesh	Fibrous material, multiple fibrous layers of polymers.	Supports CD34^+^ proliferation [60].
Biodegradable zwitterionic hydrogel	Poly-carboxybetaine acrylamide (pCBAA) hydrogel, with zwitterionic segments of 20 alternating K and E residues and a metalloproteinase-cleavable motif for degradation.	Prevents differentiation, maintains self-renewal and reduces metabolic activity of HSCs. Shows superior expansion of primitive HSCs [90].
Bio-functionalized scaffolds		
Ceramic scaffold bio-functionalized with mesenchymal cells and osteoblasts	Ceramic scaffold is cultured with hMSC and osteoblast to produce ECM and cytokines previous to HSC culture.	MSCs and osteoblasts produced a bone marrow-like environment. Functionalization increased expansion of HSCs capable of hematopoietic reconstitution [61].
Polyethylene glycol (PEG) bio-functionalized hydrogels	PEG-acrylate hydrogel was bio-functionalized by including a modified RGD peptide (involved in ECM-cell adhesion)	Supports CD34^+^ expansion and stemness better than 2D culture [48].
Bio-derived bone scaffolds (BDBS)	Scaffold from human bone is biofunctionalized with MSCs and osteoblasts.	Supports adhesion, expansion and maintenance of stemness in HSCs better than 2D co-culture [62].
Gelatin-based porous scaffold (Gelfoam) functionalized with several stromal cells	Scaffold was cultured with MSC, endothelial, osteoblasts previous to HSC on the Gel foam.	This functionalized scaffold allowed adhesion and growth of different niche cells. Supported expansion and maintenance of HSC [68].
Natural Materials		
Collagen	Elastic, biodegradable, natural component of the ECM	Co-culture in collagen supports CD34^+^ differentiation and expansion [69].
Fibrin	Natural protein, highly biocompatible.	Supports CD34^+^ adherence and proliferation [58].
Cellulose	Abundant, low-cost, non-biodegradable. Could be natural or synthetic.	Cellulose beads did not support CD34^+^ cell adhesion and proliferation [60].
Microspheres/organoids		
Collagen microspheres	MSCs were encapsulated in collagen microspheres, osteogenic differentiation was induced and subsequent decellularization to use it as scaffold for HSCs culture.	Supported mice HSC and MSC proliferation and adhesion [91].
Mesenspheres	Spheres of a low-adherence population of MSCs formed spontaneously in ultra-low adherent dishes	BM Mesenspheres support expansion of HSC [63] in co-culture.
Hematosphere	Peripheral blood mononuclear cells formed spheres in ultra-low attach surfaces.	Spheres formed from PBMNCs support extensive expansion of primitive Lin(−)CD34(+)CD38(−) HSCs [55].
Bone marrow organoid	Cord blood fibroblasts form a cellular pellet, this pellet was differentiated in vitro to a chondroid rudiment. After implantation in mice these rudiments remodeled into a functional BM niche.	The implanted organoid resembled the natural HSC niche. Host cells formed vascular structures and HSC engrafted in the organoid [92].
Decellularized ECM/tissue/organ scaffolds		
Decellularized ECM	Obtained by decellularization of ECM produced by stromal cells in vitro	Enhanced HSC adhesion and expansion of CD34^+^ cells [57].
Decellularized bovine bone marrow (DeBM)	Detergent-free decellularized bovine bone marrow with highly preserved bone marrow architecture	DeBM supported adhesion, focal localization and proliferation of mesenchymal and HSCs [18].
Decellularized porcine bone marrow	High-hydrostatic Pressurization method for decellularization of BM. Cultured with MSCs.	Supported MSC growth and differentiation. Implantation in mice induced HSC recruitment [64].
3D printing		
3D printing of MSCs-laden alginate-gelatin bioink	HSCs were cultured in a printed 3D scaffold fabricated using a mix of alginate, gelatin and MSCs.	Enhanced expansion and stemness of HSCs. Induced expression of integrins and adhesion [93].
3D printing model of endosteal and perivascular niches	3D printing of pasty calcium phosphate cement in cylinder-like format and seeded with osteogenic MSC to emulate the endosteal niche and endothelial cell laden Matrigel to mimic the perivascular niche.	Supported proliferation of CD138^+^ myeloma cells [94]

## Data Availability

Data sharing not applicable.

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
