# Peer review of "3D Scaffolds to Model the Hematopoietic Stem Cell Niche: Applications and Perspectives"

_materials, 2021, doi:10.3390/ma14030569_

Round 1

Reviewer 1 Report

This is a well written, effective review article. It summarizes the recent advances in the 3D culture systems developed to mimic the bone marrow niche. It is interesting and relevant.  The conclusion is consistent with the evidence presented in the review. 

Author Response

We thank the reviewer for her/his generous comments to our work.

Reviewer 2 Report

The authors present an overview about the hematopoietic stem cell (HSC) niches in the bone marrow (BM). The title gives the reader an idea that the paper is about “3D scaffolds to model the BM niches”, but the review focus in 3D models to mimic the niches of HSCs. I suggest replacing BM niches by HSC niches once the focus of the work is the niche of the HSC.

The work is well-structured although the text should be improved concerning some conceptual and linguistic aspects, which are listed below. I suggest accepting the review, but some aspects need to be corrected.

Abstract section

- Line 17: Replace “natural BM niches by BM microenvironment”. I suggest the authors have some attention about these concepts along the text.

  1. Introduction section

- In the second paragraph specify the names of the niches, main characteristics and where the HSCs are localized. This clarification helps to understand the quiescent and activated stage of the HSCs.

  • Lines 47 and 48: “…HSCs have been extensively studied due to their role in disease”. Specify which diseases to complement the sentence.

  • Line 76: Replace hsc by HSC.
  1. Hematopoietic stem cells and the niche

- In the subtitle above specify the niche. HSC niches or BM niches? Or the correct sentence should be “Hematopoietic stem cells and their niche”?

- Line 85: Replace “In the niche” by In their niches.

- Line 91: Replace “than” by “that”.

- Line 109: After “Schwann cells” put a “ . ” and not a “ ,”.

- There is a lack of information about the HSC niches (only the last paragraph of the topic mentions the names). Specify in the second paragraph of this section the existence of different HSC niches to facilitate the reading of the next paragraphs.

  1. Applications of 3D models of BM niche for cell culture

- Lines 206, 207 and 208 (ref. 69): Please rewrite the sentence, it is confusing.

  1. Current 3D models of the bone marrow niche

- Line 253: Replace “that will allow cell infiltration” by “which allow cell infiltration”.

- Line 255: Replace “for” by “of”.

It could be interesting to mention in the last paragraph that recently some scaffolds are maintained under different conditions (static, dynamic, normoxia and hypoxia) aiming to recapitulate as close as possible the native BM microenvironment and the HSC niches.

4.2.2 Microspheroids/Organoids

- Lines 362 -364 (ref 54): Please rewrite the sentence, it is confusing.

- Line 391: The authors wanted to say “proliferation of HSC” instead of “proliferation of MSC”?

  1. Conclusions

- Lines 403 and 404: I suggest rewriting the sentence: “In previous sections we have discussed the complexity of the bone marrow niches and the elements of the niche relevant to HSC physiology”.

Author Response

We thank the reviewer for the feedback and the opportunity to improve our manuscript.

All minor corrections suggested have been implemented and highlighted in red in the manuscript.

Comment: It could be interesting to mention in the last paragraph that recently some scaffolds are maintained under different conditions (static, dynamic, normoxia and hypoxia) aiming to recapitulate as close as possible the native BM microenvironment and the HSC niches.

Response: We included some information regarding these conditions in section 4. Also highlighted in red.

Reviewer 3 Report

The article by Congrains et al reviews the scaffold options for growth, expansion, tissue engineering and studying of hematopoietic stem cells (HSC). The work first reviews the function of HSC, then lists the needs for 3D culturig of HSC, and finally reviews the available scaffold materials and their production methods suitable for these specific cells.

I have several comments for the improvement of this manuscript:

  1. At several instances when the authors refer to published literature, they are too vague on on what has actually been reported as results. Examples include: line 58-59, line 171-174, line 191-193, line 232-235 and elsewhere as well. Should re-check the text keeping in mind that results or exact studied methods are stated, not only refering to "several applications" or a general "scaffold".
  2. Writing the abbreviations in full the first time when using them, such as "CXCL12" and "HAL".
  3. A bit more explanation on the effect of stiffness on the HSC behavior in the niche. There apparently is a stiffness gradient from endosteal niche to the central marrow, but it would be interesting to hear more about this and has stiffness gradient already been used in any scaffold structures?
  4.  For ex vivo expansion of cells, the recovery of cells from the scaffold is critical and so far has poor success rate. This issue and potential solutions should be briefly discussed in chapter 3.1 and perhaps also acknowledged in later chapters as well.
  5. Barrier properties of tissues or scaffolds should also be mentioned under 3.3. when developing drug testing platforms.
  6. For HTS assays, imaging is mentioned, but also problems and solutions for PCR assays combined with HTS should be mentioned.
  7. Even though the focus of the review as well as the journal is on materials, scaffold-free systems should be mentioned. Examples include hanging-drop method and suspension cultures in rotating bioreactors. Have these been used for HSC applications?
  8. Decellularization of bone marrow niche is in focus, but the methodology of decellularization could be more carefully explained. References for the harmfulness of enzyme and detergent residues present after decellularization. And why are these chemicals used, if freeze-thaw cycles only can also yield a cell-free, well functioning scaffold? How is the decellularization process different, when not using chemicals at all? A schematic figure might also be good for clarity.
  9. Are Figures 2 & 3 published here for first time or are they taken from the referenced articles? Figure caption should indicate this, if figure has been published previously.
  10. About Table1: The production method for synthetic scaffolds from PLGA, PCL and PU are not mentioned, but should be included. Further, are the zwitterionic pCBAA  and PEG the only hydrogels that has been studied for HSC applications? And which ceramic material is used for HSC application (row 9)?
  11. The reference listing could use a proof-reading, but this might be part of the editorial process anyways. Some of the used review references are a bit old (for example: Clinical applications of mesenchymal stem cells 2012) for this fast moving field, but this is minor issue.

Author Response

  1. At several instances when the authors refer to published literature, they are too vague on on what has actually been reported as results. Examples include: line 58-59, line 171-174, line 191-193, line 232-235 and elsewhere as well. Should re-check the text keeping in mind that results or exact studied methods are stated, not only refering to "several applications" or a general "scaffold".

We thank the reviewer for his/her feedback. In some instances, the results and details of some of the references are mentioned in other sections of the manuscript; however, we found some cases in which information was missing and we included additional information.

  1. Writing the abbreviations in full the first time when using them, such as "CXCL12" and "HAL".

We corrected this problem

  1. A bit more explanation on the effect of stiffness on the HSC behavior in the niche. There apparently is a stiffness gradient from endosteal niche to the central marrow, but it would be interesting to hear more about this and has stiffness gradient already been used in any scaffold structures?

We thank the reviewer for the suggestion. We included new information regarding the effect of the stiffness in HSC behavior and potential role in the scaffold in section 2.

  1. For ex vivo expansion of cells, the recovery of cells from the scaffold is critical and so far has poor success rate. This issue and potential solutions should be briefly discussed in chapter 3.1 and perhaps also acknowledged in later chapters as well.

We thank the reviewer for pointing out this important issue. We discussed the problem of cell recovery and we proposed some alternatives in the revised version.

  1. Barrier properties of tissues or scaffolds should also be mentioned under 3.3. when developing drug testing platforms.

We discussed briefly the issue in section 3.3 of the revised version

  1. For HTS assays, imaging is mentioned, but also problems and solutions for PCR assays combined with HTS should be mentioned.

We are including additional information and a reference regarding gene expression based- HTS in section 3.3.

  1. Even though the focus of the review as well as the journal is on materials, scaffold-free systems should be mentioned. Examples include hanging-drop method and suspension cultures in rotating bioreactors. Have these been used for HSC applications?

We are including a hanging-drop system and some references of bioreactor models used for HSC culture in section 4. In addition, other scaffold free systems are mentioned in section 4.2.2.

  1. Decellularization of bone marrow niche is in focus, but the methodology of decellularization could be more carefully explained. References for the harmfulness of enzyme and detergent residues present after decellularization. And why are these chemicals used, if freeze-thaw cycles only can also yield a cell-free, well functioning scaffold? How is the decellularization process different, when not using chemicals at all? A schematic figure might also be good for clarity.

We thank the reviewer for the opportunity to improve the manuscript and clarify the methodology for the readers. We included a graphical summary (figure 2 in the revised version), the details have been published in Bianco et al. ; however, we made the reference clearer in the revised version. We included additional information about the effect of detergents in the decellularization process and some references.

  1. Are Figures 2 & 3 published here for first time or are they taken from the referenced articles? Figure caption should indicate this, if figure has been published previously.

All the images in this review would be published for the first time. The SEM images are part of the characterization of the DeBM, however, these particular images were never published. We clarified it the legend.

  1. About Table1: The production method for synthetic scaffolds from PLGA, PCL and PU are not mentioned, but should be included. Further, are the zwitterionic pCBAA and PEG the only hydrogels that has been studied for HSC applications? And which ceramic material is used for HSC application (row 9)?

We included additional information about these synthetic scaffolds, results and procedures, in section 4.2.1. We added information about the zwitterionic gels, but we could not find details regarding the type of ceramic used by Bourgine et al. in the article.

  1. The reference listing could use a proof-reading, but this might be part of the editorial process anyways. Some of the used review references are a bit old (for example: Clinical applications of mesenchymal stem cells 2012) for this fast moving field, but this is minor issue.

We will keep in mind the reviewer’s suggestion during the editorial process of the manuscript to avoid errors in the reference list.

Round 2

Reviewer 3 Report

The revised manuscript by Congrains et al has provided good answer for the review questions. Two minor notes for improvement could be taken into account and then the manuscript is ready for publication:

  1. In chapter 3.3. Matrigel is discussed right after synthetic polymers. This can be now falsely understood that Materigel would be synthetic polymer, which it is not.
  2. In chapter 4.2.1 could be reference to Table 1.

Author Response

1. In chapter 3.3. Matrigel is discussed right after synthetic polymers. This can be now falsely understood that Materigel would be synthetic polymer, which it is not.

We thank the reviewer for the opportunity to correct this mistake. We re-wrote the sentence and changes were highlighted in blue

2. In chapter 4.2.1 could be reference to Table 1.

We referenced the table in section 4.2.1 in the revised version.

We thank the reviewer and the editors for their contribution to improve the manuscript.